# Barriers to Access the Pap Smear Test for Cervical Cancer Screening in Rural Riverside Populations Covered by a Fluvial Primary Healthcare Team in the Amazon

**DOI:** 10.3390/ijerph19074193

**Published:** 2022-04-01

**Authors:** Débora C. B. da Silva, Luiza Garnelo, Fernando J. Herkrath

**Affiliations:** 1Instituto Leônidas e Maria Deane, Fundação Oswaldo Cruz, Rua Teresina 476, Manaus 69057-070, Brazil; debora.enf.ufam@gmail.com (D.C.B.d.S.); luiza.garnelo@fiocruz.br (L.G.); 2Escola Superior de Ciências da Saúde, Universidade do Estado do Amazonas, Av. Carvalho Leal 1777, Manaus 69065-001, Brazil

**Keywords:** uterine cervical neoplasms, cancer screening tests, rural population, access to health services

## Abstract

Cervical cancer is a major public health problem, especially in the north region of Brazil. The aim of the study was to identify the factors associated with not undergoing the cervical cancer screening test in rural riverside populations in the Amazon. A cross-sectional home-based survey was carried out in 38 locations covered by a fluvial primary healthcare team, and the administrative records of the screening tests from January 2016 to May 2019 were analyzed. After the descriptive analysis, logistic regression was performed considering the outcome of having undergone cervical cancer screening within the past three years. Of the 221 women assessed, 8.1% had never undergone the test, and 7.7% had undergone it more than three years ago. Multiparity (OR = 0.76 (95%CI = 0.64–0.90)), occupation in domestic activities (OR = 0.31 (95%CI = 0.11–0.89)), and lack of knowledge of the healthcare unit responsible for the service (OR = 0.18 (95%CI = 0.04–0.97)) were associated with not undergoing the cervical cancer screening test. The administrative records revealed that the screening test was performed outside the recommended age range (24%), performed needlessly (9.6%) with undue repetitions (3.2%), and a high percentage of the samples collected were unsatisfactory (23.5%). The findings revealed the existence of barriers for riverside women to access cervical cancer screening tests.

## 1. Introduction

Cervical cancer is a public health problem, and women who have more difficulty accessing health services are more affected [1]. Although it is a preventable and potentially curable disease, cervical cancer is one of the most common causes of female cancer mortality worldwide [2].

Globally, more than 500,000 women are diagnosed each year, and the disease caused the death of 311,000 women in 2018, with a higher occurrence in less developed countries [3]. Cervical cancer ranks second in incidence and mortality in low human development countries [4]. By 2030, there will be an increase in the annual number of new diagnoses of cervical cancer, and the estimated number of deaths will be 400,000 women worldwide [2]. According to the National Cancer Institute, the estimated gross rate of cervical cancer in Brazil was 16.35/100,000 in 2020. The estimated gross rate in the state of Amazonas in the same year was 33.8/100,000, the highest in the country [5].

Routine screening is an effective method for detecting precancerous lesions. However, there is still a high rate of deaths in women from countries with poor access to early detection and treatment services. Furthermore, detection in less developed countries is also hindered by competing health priorities, insufficient financial resources, and deficiencies in health systems [6].

In Brazil, the guidelines for cervical cancer screening recommend an annual screening test (Pap smear test) for women aged 25–64 years, and an interval of three years after two consecutive negative tests [7]. Several difficulties for performing the screening test are pointed out in the literature, such as access to health services, shame, fear, beliefs, taboos, and low education, and limited health promotion and education [8].

Residents of rural areas face major barriers to access health services that meet their needs [9]. In Brazil, assessments have shown that the provision and the delivery of the quality of healthcare services in rural populations are fragmented and discontinuous [9,10,11]. The most recent version of the National Primary Healthcare Policy in Brazil [12] sought to increase access to health care using fluvial primary healthcare teams. These fluvial health teams (FHT) provide itinerant primary care for populations in remote locations along rivers. The FHT are an alternative to increase healthcare coverage, but they face limitations due to the short time of stay in each location [13]. Thus, it is imperative to understand the potentialities and limitations of the FHT to provide routine healthcare services, such as cervical cancer screening.

Studies investigating the access to healthcare services in the rural areas of the Amazon are scarce [9]. An important tool for planning public policies to control cervical cancer is to know what prevents women from undergoing the Pap smear test [14]. Thus, the aim of the study was to identify the factors associated with the non-performance of cervical cancer screening in rural riverside populations along the Rio Negro River, Manaus, Amazonas, covered by the FHT.

## 2. Methods

The study is a cross-sectional household-based survey carried out in 38 rural riverside locations along the Rio Negro River from the rural area of Manaus to the municipality of Novo Airão, Amazonas, Brazil (Figure 1). The administrative records of Pap smears performed in the territory from January 2016 to May 2019 were also analyzed in the study. The study population was distributed in five micro-areas covered by an FHT, whose activities have been described elsewhere [10].

The study was part of a larger project that described the socioeconomic characteristics, health conditions, and use of and access to health services of the rural Amazonian population residing along the Rio Negro River. Stratified random sampling was performed based on the number of individuals and households in each community, as reported by the community health workers (CHW), totaling 2342 people from 765 households, including 466 women from the target groups of this study. The sample size calculation considered the representativeness of the groups of interest in the larger project, and the probability of finding individuals from each group within each household. The calculation considered a prevalence of 50% of the outcomes of interest at 5% precision and 10% of possible losses or refusals, which were adjusted for the finite population, resulting in 239 households. The aim was to reach 221 people of the study population: adult women aged 18–59 years, and women under the age of 18 who had children under the age of two years or were pregnant.

The interview was carried out using a structured questionnaire containing seven themes developed in the Research Electronic Data Capture (REDCap) software. The questionnaire was administered directly to the resident by trained interviewers. The questions of specific interest for this study were related to women’s health, and were designed to investigate four specific variables related to cervical cancer screening tests (last time test was performed; main reason for never undergoing the test; location of the last test; and delivery time of the result of the last test). Variables related to geographic characteristics, socioeconomic conditions, and those related to healthcare services were also considered, as they can potentially explain why the test was not performed. The pilot study was carried out in rural areas of Manaus that were not included in the main study. The present study was based on the theoretical conceptual model proposed by the World Health Organization [15], covering structural and intermediary determinants of health inequities related to the outcome of interest.

The Municipal Health Department provided the administrative records of the Pap tests performed from January 2016 to May 2019 in spreadsheets containing the identification of women, date of collection, types of cells in the sample, test results, and time to deliver results.

A descriptive analysis of the data was performed, including the four variables related to the outcome of interest. Descriptive analyzes were also performed according to the age groups < 25 years and 25–59 years. Data on the accomplishment of screening test were compared with the administrative records to identify possible inconsistencies. The number of screening tests per location and the pattern of repetition within the period were described. Then, a bivariate logistic regression analysis was performed, considering the dichotomous outcome of the performance or non-performance of cervical cancer screening over the past three years for the age group from 25–59 years and the independent variables, using the data from the household survey. Variables with *p* ≤ 0.20 in the bivariate analyses were included in the multiple hierarchical analysis, considering the structural and intermediary determinants according to the theoretical conceptual model. Variables with *p* ≤ 0.10 were retained in the final model. The analyses were carried out using the IBM SPSS Statistics program, version 22.0.

The study was approved by the Research Ethics Committee (protocol CAAE nº 57706316.9.0000.0005). Participants signed an informed consent to participate in the study.

## 3. Results

In the study, 221 women were assessed. The mean age was 35 years (±SD = 11.04). A total of 20.8% (*n* = 46) of the women were under the age of 25 years, and twelve women were under the age of 18 years and had two-year old children or were pregnant. As for multiparity, the mean number of children was 3.85 (±SD = 2.68). A total of 8.1% of women reported they had never undergone the Pap smear test, and 7.7% reported they had undergone the Pap test more than three years ago. Among the women aged 25 years or older who had never undergone the test, they reported the following reasons: did not deem it necessary and felt ashamed. Most of the women received the test results within three months of the test, and stated that their last test was performed at a public health service (Table 1).

Of the women assessed, 136 women had one to nine years of education (61.5%). The main occupations were domestic activities (46.2%) and agriculture/extractive activities/fish farming (35.7%). The average monthly family income was R$908.58, which was lower than the minimum wage at the time (R$998.00, equivalent to US$263.57). Regarding self-perception of health, most women considered their health status as very good or good (54.8%). When asked which healthcare unit was responsible for covering their community, 40.7% reported it was the “boat” of the Municipal Health Department (referring to the FHT), and that their last appointment at the unit was scheduled through the CHW (53.4%). Most women classified the degree of difficulty scheduling an appointment at the unit as “easy” or “very easy” (49.1%), but a significant percentage reported having difficulty scheduling an appointment (28%). When these women were sick or in need of health care, they firstly went to the primary healthcare unit (57.5%), and the main means of transportation to the service were on foot (34.4%) or by boat/canoe (57.5%). The women considered that the distance from their home to the healthcare unit was, in general, close and easily accessible (55.7%) (Table 2).

There was a difference in the travelling time to the healthcare unit during the flood and drought periods. The average time spent during the drought period was of approximately 29 min, more than twice the average time spent in the flood season. The measurements from georeferenced coordinates showed an average distance of 2.40 km (±SD = 3.47) from the household to the docking locations of the fluvial health mobile unit, ranging from 25 m to 15.50 km. The average distance from the household to the primary healthcare unit in the territory was 6.89 km (±SD = 6.47), ranging from 14 m to 22.10 km (Table 2).

Table 2 shows the description of the structural and intermediary independent variables used in the study, as well as the results of the bivariate analyses between the variables and the outcome of not performing the preventive screening test over the past three years for women aged 25–59 years. The older the age (OR = 0.94 (95%CI = 0.89–0.99)) and the greater the number of birth deliveries (OR = 0.80 (95%CI = 0.69–0.94)), the less the chance of having undergone the test in the past three years.

In the multiple hierarchical regression analysis, domestic activities (OR = 0.31 (95%CI = 0.11–0.89)), multiparity (OR = 0.76 (95%CI = 0.64–0.90)), and lack of knowledge of a primary healthcare unit in the territory were associated with the outcome (OR = 0.18 (95%CI = 0.04–0.97)). Women who had more children, who reported domestic activities, and who were unable to inform or were unaware of the existence of a healthcare unit responsible for the community were less likely to have undergone a preventive screening test in the past three years (Table 3).

Administrative records showed that within the period assessed, 651 women were examined, and 1097 screening tests were performed (303 tests in 2016, 295 tests in 2017, 269 tests in 2018, and 230 tests in 2019). The mean age of the women examined was 36 years (±SD = 13.29), ranging from 14 to 83 years. Of these records, 24% were related to women outside the age group recommended by the Brazilian guidelines (25–64 years old) for cervical cancer screening, 35 women over 64 years old (3.2%), and 229 women were aged 24 years or younger (20.8%). In relation to the total number of screening tests, 35 (3.2%) were inappropriately repeated (two screening tests in the same year, with no indication for repetition). A total of 105 screening tests (9.6%) were performed unnecessarily, considering that the woman had presented negative results for neoplasia in the two previous years, which would indicate an interval of three years before performing a new screening test. The joint assessment of these situations indicates that 140 screening tests (12.8%) should not have been performed (Table 4).

On the other hand, the screening tests of 210 (19.1%) women were potentially delayed, that is, they had not undergone the test for more than three years. However, when comparing administrative data with self-reported data (collected in the survey), 26 homonyms were found, and of these, only two women reported not undergoing the test for more than three years. Of the samples of epithelium cells found in the results of the screening tests, 102 samples (9.3%) had no information, considering that some screening tests had just been collected, and the results were not available; 258 samples (23.5%) had insufficient cells (squamous or glandular epithelium found separately), indicating the need to repeat the screening test; 601 samples (54.8%) had squamous/glandular epithelium; 34 samples (3.1%) had squamous/metaplastic epithelium; and 102 samples (9.3%) had squamous/glandular/metaplastic epithelium (Table 4).

## 4. Discussion

Most of the women assessed in the study reported having been screened for cervical cancer less than a year ago, with a satisfactory delivery time of the test results. Among women in the recommended age group for the test, occupation, multiparity, and lack of knowledge about the healthcare unit responsible for the care were associated with not performing the test over the past three years. Administrative records showed a high percentage of unsatisfactory samples collected and tests performed outside the age group recommended by the national guidelines. This problem was associated with limited staff qualification, and gynecological complications common in younger age groups [16,17].

The coverage of the Pap smear test provided by the FHT (84.2%) is similar to that found in national population-based studies, such as that reported in a survey carried out in capital cities that showed a similar percentage of performance of the screening test in an interval of up to three years [14]. Findings from the Primary Healthcare Access and Quality Improvement Program (PMAQ-AB) [18] showed higher coverage for the entire primary care network in the country (93.3%). However, the findings revealed that the percentage in the north region was 84.1%. The north region of Brazil has shown low coverage of cervical cancer screening in national household surveys (75.5%) [19], with slightly better results in capital cities (85.3% in Rio Branco and 85.7% in Boa Vista) [20,21], which is slightly higher than that found in the rural population in the present study covered by the FHT. In contrast, the lower percentage of delays in performing tests in the population covered by the fluvial team (7.7%) was lower than the 11.2% recorded for the North region in the PMAQ-AB survey [18].

The percentage of women surveyed who had never undergone the Pap smear test was 8.1%. However, this number decreases to 2.9% when including the women in the age group prioritized by technical guidelines. This percentage is much lower than that found in the aforementioned PMAQ-AB study (6.7% for the whole country and 7.8% for the North region), as well as in another riverside location [22], which revealed a worrying percentage (70%) of women who had never undergone the screening test. The coverage of primary care provided by the FHT in the territory may have contributed favorably to this result. Despite the reduced number of deaths from cervical cancer in Brazil, the same did not occur in the north of the country. This suggests that early detection, especially in areas of greater vulnerability and risk [23], that is, in the rural areas, must be increased.

Failure to undergo the screening test may be related to personal issues, such as fear and shame, but lack of knowledge about the importance of the test [24,25] and unwillingness to perform it has contributed to non-performance. A study in rural communities in Nigeria showed the effect of health education in raising the level of women’s knowledge of cervical cancer, which led to an increase in screening tests from 2% to 70.5% [26]. The feeling of fear and shame when undergoing the Pap smear test, due to the lithotomy position, reported by the riverside women in the survey is recurrent in several other studies [27,28,29,30]. Fear of a positive result, absence of symptoms, low education and income, and lack of time were also important obstacles for seeking healthcare services [14,31].

Although education [32], income, and moral and affective values also interfere in the perception of risk and preventive practices [33], the organization of healthcare services and actions of health professionals [1,18,27] are significant aspects to obtain satisfactory results. Difficulties in accessing the healthcare unit, negligence of professionals in providing humanized care, lack of instructions and information offered to these women about cancer, and the importance of the screening test are closely associated with the failure to undergo the Pap smear test [14,31,34,35,36]. The precariousness in the structure and environment of healthcare units for performing the Pap smear test [18] are associated with limited scheduling time for the test, low flexibility in scheduling appointments, delay in delivering results, and barriers to access specialized services [16,27,37]. All these aspects require improvement when planning actions for cervical cancer prevention in the north region. Managers and health professionals must develop strategies to minimize the interference caused by factors that lead to the non-performance of the Pap test [38], and the unfavorable conditions of availability and access to health services in rural areas [9,39].

The overload of work related to the combination of agriculture/extractive activities/fishing and domestic tasks can make it difficult for riverine women to adhere to preventive behaviors. This difficulty is caused by the hard and long working hours of the women, the limited opening hours of the healthcare unit, when appointments occur on specific days in the month, as well as delay in attendance [40,41].

Although most women in the survey claimed to know which healthcare unit was responsible for covering their community, 11.3% of the women were unaware which primary healthcare unit was responsible for their territory. This lack of knowledge was associated with not undergoing the Pap smear test in the adjusted analysis. Not knowing which healthcare unit is responsible for the territory suggests a fragility in ties, which is essential for establishing relationships of affection and trust between the patients and healthcare workers. In addition to the therapeutic potential of ties [12], trust in the healthcare workers is an essential element for outcome of this study, as they encourage women to participate in self-care and health promotion activities [42]. The itinerant healthcare model of the FHT makes bonding difficult, given the limited permanence of the fluvial health unit in each location. However, the relatively low percentages of non-performance of the screening test and recognition of the performance of FHT suggest that the teams have been surmounting the inherent limitations of their work.

One of the challenges found in the study was accessing riverside communities, since overcoming distances in the Amazon implies not only traveling kilometers, but also having time and conditions for river travel. The previously mentioned [29] influence of the geographical barriers on carrying out the Pap tests in rural areas is worsened by the lack of regular public transportation and limited financial resources of the population to pay for transport. These conditions oblige women to walk or use school transport to access healthcare services.

Although the interviewees consider the distance between the healthcare units and their house to be short and easy to access, this interpretation needs to be taken with reservation. Due to the characteristics of Amazonian life, what the residents—accustomed to the great geographical barriers—consider to be “close” and “easy” routes can be seen as dreadful journeys in the eyes of outsiders when compared with the conditions in the urban areas. The difficulty of access is increased by the seasonal rhythm of the rivers, which increased by 2.3 times the average travelling time to the healthcare unit. These results corroborate the findings of a study that described the performance of a fluvial healthcare unit that reported that the perception of distance and time in the Amazon are altered by the type of transport, engine power, and seasonality of the river (flood or ebb) [13].

The positive results obtained by the fluvial primary healthcare team responsible for the territory also suggest the protagonism of CHW, who are in permanent contact with the population, in contrast to the itinerant work of the rest of FHT. The CHW, who are members of the community, are key to offering preventive care for cervical cancer, as they have intra- and inter-family ties [29] that favor longitudinal care and reduce access barriers. Furthermore, they identify and search for the women whose Pap smear test results show any abnormalities, or who have missed appointments [10]. As they are in closer contact with the women and families, these professionals can identify places and women who need care, and actively search for assistance for them [39]. CHW can also promote health education. The increase in knowledge about the disease, the test, and its purpose is able to raise the rate of performance of the test [43].

In addition to multiparity, which has a negative influence on screening [35,44], it is difficult for the women to seek the health services when they are primarily responsible for the home and childcare. Thus, the health teams must adopt alternative measures to actively search for the women, and encourage adherence to preventive measures [41]. Among these measures, the literature suggests taking advantage of the gestational and puerperium periods to increase the chances of women maintaining contact with health services, and undergo periodic examinations [44]. However, Ribeiro et al. [45] concluded that contact with the health service for prenatal care was not a determining variable to ensure access to the screening test, thus indicating a loss of opportunity for screening. The literature also points out that multiparous women, who could benefit most from cervical cancer screening during antenatal care, are the ones who least undergo it [46].

Administrative data from the FHT revealed that approximately one fifth of the tests were performed on women under the age of 25 years, evidencing a concentration of efforts in an age group in which cancer rarely manifests. The regulations in force in Brazil acknowledge young age groups as the starting point for cervical cancer screening, but do not recommend priority screening for this age group [7]. The data also revealed that a few tests were performed by the FHT in women older than the recommended age. These findings are similar to those found in other studies [17,37,47], suggesting that the profile of the results achieved at the FHT is compatible with the overall performance of the primary health care. Some women were submitted to the test at a higher frequency than recommended by the national guidelines. In addition to wasting time and resources, performing unnecessary procedures can contribute to limiting access for women with delayed screening tests.

The cervix material collection in Pap smears aims to gather cellular elements representative from the most common sites of cervical cancer. The retrieval of metaplastic or endocervical cells from the squamocolumnar junction is considered an indicator for the quality of smears. It is imperative that health professionals be qualified to ensure that this cellular material is properly obtained at the squamocolumnar junction. If not, they will not provide the woman with all the benefits of cervical cancer prevention [7].

In the present study, almost one quarter of the samples contained only one cell type (squamous or glandular), which is not a desirable quality indicator. This percentage is much higher than the recommended 5%, often requiring the repetition of the screening test [48,49]. The percentage of unsatisfactory quality smears performed at the FHT was higher than the percentage of unsatisfactory cytopathological samples (about 5%) in other municipalities in the state of Amazonas [50]. In addition to the increase in costs, and the loss of opportunity for women to adhere to the test, commitment to the quality of the collection indicates the need for the training and qualification of professionals to optimize resources, and reach goals established by the services [47].

In 2020, the World Health Organization proposed a global strategy to accelerate the elimination of cervical cancer as a public health problem through cost-effective, evidence-based interventions. This global strategy states that all countries must reach and maintain an incidence rate below 4 per 100,000 women. It also defines the 90–70–90 targets which must be met by countries by 2030: 90% of girls fully vaccinated with HPV vaccine by age 15, 70% of women screened with a high-performance test at age 35 and again at age 45, and 90% of women identified with cervical disease receiving treatment [2]. Thus, efforts should include focus on achieving adequate vaccination coverage. Although high in most micro-regions of the country, vaccination coverage faces problems in the North region of Brazil, particularly in the state of Amazonas [51].

The transition to HPV testing as the primary method of screening for cervical cancer has also been recommended, considering the higher level of the test performance and the difficulties faced by low- and middle-income countries in conducting cytology-based programs [2,52]. Self-sampling HPV nucleic acid testing is an additional option that has been shown to be as effective as clinical-collected samples [53], and with great potential to contribute to increasing cervical cancer screening coverage [54], and to overcoming barriers to access the health services for early diagnosis in areas with lower availability, such as rural riverside localities [52,55].

Some limitations of this study must be considered. Causal inferences should be carefully evaluated, considering the cross-sectional design of the study. Another limitation is related to the difficulty in linking data from administrative records with the data obtained in the survey. Moreover, as the information on the Pap test was self-reported in the survey, it is subject to information bias. On the other hand, data from a representative sample for the study population were analyzed, based on a robust theoretical-conceptual model, and the administrative data covered the universe of screening tests carried out in the evaluated period.

## 5. Conclusions

Although a good performance of the FHT was identified, with a satisfactory coverage of cervical cancer screening in the studied population, the findings showed that there are barriers for women in rural riverside locations to access the screening test, including organizational barriers. The results also suggest that healthcare teams face difficulties in adequately managed care, which was evidenced by the possible delays in performing the test, and the significant number of unsatisfactory samples. These findings can help the development of actions related to service organization strategies, retraining of professionals to perform quality tests, transitioning to other testing strategies, and identification and active search for women who do have not access to the service to undergo periodic screening tests as recommended by the technical guidelines. The barriers identified are a risk for the non-identification of primary lesions for the development of cervical cancer. Furthermore, the findings reinforce the importance of regular health promotion actions, which can increase the ties between the women and the health teams, and, among other favorable health outcomes, promote the performance of the cervical cancer screening following the periodicity recommended.

## Figures and Tables

**Figure 1 ijerph-19-04193-f001:**
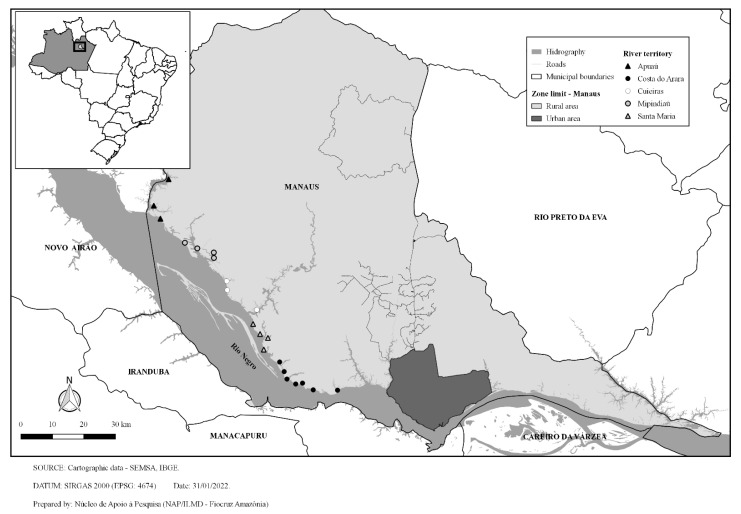
Riverside rural locations covered by the study.

**Table 1 ijerph-19-04193-t001:** Variables related to cervical cancer screening (*n* = 221).

Variable	Total Sample *n* (%)	<25 Years *n* (%)	25–59 Years *n* (%)
**Last screening test**			
Never	18 (8.1)	13 (28.9)	5 (2.9)
Less than 1 year ago	150 (67.9)	25 (55.5)	125 (71.4)
1 -| 2 years ago	26 (11.8)	6 (13.3)	20 (11.4)
2 -| 3 years ago	10 (4.5)	1 (2.2)	9 (5.1)
More than 3 years ago	17 (7.7)	1 (2.2)	16 (9.1)
**Reasons for not performing screening test (*n* = 18)**			
Never had intercourse	1 (5.6)	1 (7.7)	-
Did not deem necessary	5 (27.8)	1 (7.7)	4 (80.0)
Felt ashamed	3 (16.7)	2 (15.4)	1 (20.0)
Had never been instructed on undergoing the test	1 (5.6)	1 (7.7)	-
Had trouble scheduling the test	2 (11.1)	2 (15.4)	-
Because of age	3 (16.7)	3 (23.1)	-
Fear	2 (11.1)	2 (15.4)	-
Could not inform	1 (5.6)	1 (7.7)	-
**Result waiting time (*n* = 203)**			
Less than a month	15 (7.4)	2 (6.2)	13 (7.6)
1 -| 3 months	113 (55.7)	20 (62.5)	92 (54.1)
3 -| 6 months	38 (18.7)	4 (12.5)	34 (20.0)
Six months or more	1 (0.5)	-	1 (0.6)
Has not received the result yet	32 (15.8)	-	26 (15.3)
Never received the result	3 (1.5)	6 (18.8)	3 (1.8)
Never went to get the result	1 (0.5)	-	1 (0.6)
**Location of screening test (*n* = 203)**			
Public service	194 (95.6)	31 (96.9)	162 (95.3)
Private service	8 (3.9)	1 (3.1)	7 (4.1)
Does not know/did not answer	1 (0.5)	-	1 (0.6)

**Table 2 ijerph-19-04193-t002:** Description of independent variables and unadjusted odds ratios for not performing cervical screening in the past three years in women aged 25 to 59 years.

Variable	Mean (±SD)/ *n* (%)	OR	95% CI	*p*-Value
**Structural determinants**				
*Age*	35 (±11.04)	0.94	0.89–0.99	0.010 *
*Years of education*		1.13	0.99–1.30	0.061 ^a^
Never went to school	4 (1.8%)			
1–9 years of education	136 (61.5%)			
10–12 years of education	70 (31.7%)			
13 years of education of more	9 (4.1%)			
No information	3 (1.4%)			
*Family income (R$)*	908.6 (±769.0)	1.00	1.00–1.00	0.452
*Occupation*				
Others	119 (53.8%)	ref		
Housewife	102 (46.2%)	0.36	0.14–0.93	0.034 *
*Access to care*				
Easy		ref		
Difficult		1.15	0.36–3.76	0.812
**Intermediary determinants**				
*Self-perception of general health*				
Very good/good	121 (54.8%)	1.24	0.25–6.21	0.797
Regular	71 (32.1%)	1.26	0.23–6.85	0.788
Poor/very poor	15 (6.8%)	ref		
No information	14 (6.3%)			
*Level of difficulty to schedule appointments*		1.35	0.82–2.25	0.234
Very easy/easy	86 (49.1%)	1.06	0.36–3.12	0.915
Not easy nor difficult	29 (16.6%)	3.91	0.45–34.21	0.218
Difficult/very difficult	49 (28%)	ref		
Does not know/did not answer	11 (6.3%)			
*First place you go when you are sick or in need of health care*				
Primary care	162 (73.4%)	1.01	0.32–3.24	0.981
Another healthcare service	35 (15.8%)	ref		
None/another/no information	24 (10.9%)			
*How do you travel to get to the service you usually look for?*				
On foot	76 (34.4%)	ref		
Boat/canoe	127 (57.5%)	0.57	0.21–1.56	0.277
Other/no information	18 (8.2%)			
*Knowledge of the healthcare unit responsible for covering the community*				
Community health clinic or boat	170 (76.9%)	ref		
Healthcare unit outside the community	26 (11.8%)	0.98	0.21–4.69	0.984
There is not one/does not know	25 (11.3%)	0.26	0.06–1.10	0.067 ^a^
*Time it takes from home to the healthcare unit during floods*	15.44 (±15.44)	1.00	0.97–1.03	0.861
*Time it takes from home to the healthcare unit during drought*	29.19 (±32.40)	1.00	0.98–1.01	0.887
*Opinion about the distance from home to the health unit*				
Close	141 (63.8%)	ref		
Far	54 (24.4%)	1.78	0.49–6.52	0.385
No information	26 (11.8%)			
*How did you schedule the last appointment at the healthcare unit?*				
Scheduled or through the team	140 (63.4%)	ref		
On their own	55 (24.9%)	1.76	0.48–6.46	0.393
No information	26 (11.8%)			
*Waiting time on the last appointment*	85.25 (±81.3)	1.00	1.00–1.00	0.765
*Distance from the house to the boat stop*	2.4 (±3.4)	0.99	0.88–1.12	0.885
*Distance from home to the health unit*	6.8 (±6.4)	0.98	0.92–1.05	0.607
*Multiparity*	3.85 (±2.68)	0.80	0.69–0.94	0.005 **

* *p* < 0.05; ** *p* < 0.01; ^a^ Variables with *p* < 0.20 were included in the multiple analysis.

**Table 3 ijerph-19-04193-t003:** Adjusted odds ratios for performing cervical screening in the past three years in women aged 25 to 59 years.

Variable	Model 1		Model 2		Model 3	
	OR (95% CI)	*p*-Value	OR (95% CI)	*p*-Value	OR (95% CI)	*p*-Value
**Structural determinants**						
*Age*	0.93 (0.89–0.98)	0.006 **	0.96 (0.90–1.02)	0.159		
*Years of education*	0.97 (0.92–1.02)	0.187				
*Occupation (ref. others)*						
Housewife	0.36 (0.14–0.94)	0.037 *	0.30 (0.10–0.88)	0.028 *	0.31 (0.11–0.89)	0.029 *
**Intermediary d** **eterminants**						
*Knowledge about the healthcare unit responsible for covering the community (ref. in the community)*						
Healthcare unit outside the community			0.84 (0.16–4.30)	0.835	0.86 (0.17–4.37)	0.857
There is not/does not know			0.20 (0.04–1.01)	0.051	0.18 (0.04–0.97)	0.045 *
*Multiparity*			0.81 (0.67–0.96)	0.018 *	0.76 (0.64–0.90)	0.002 **

* *p* < 0.05; ** *p* < 0.01.

**Table 4 ijerph-19-04193-t004:** Administrative data on Pap smear tests performed on the study population from January 2016 to May 2019, types of cells, and results by age group.

Variable	Total Sample 14–83 Years *n* (%)	<25 Years *n* (%)	25–59 Years *n* (%)
**Total of screening tests (*n* = 1097)**	1097 (100)	229 (100)	803 (100)
Inappropriately repeated	35 (3.2)	7 (3.1)	26 (3.2)
Unnecessary performed	105 (9.6)	11 (4.8)	87 (10.8)
Potentially delayed	210 (19.1)	45 (19.7)	146 (18.2)
**Types of epithelium cells (*n* = 1097)**			
No information	102 (9.3)	17 (7.4)	85 (10.2)
Squamous epithelium	257 (23.4)	54 (23.6)	193 (23.1)
Glandular epithelium	1 (0.1)	1 (0.4)	-
Squamous/glandular epithelium	601 (54.8)	137 (59.8)	451 (53.9)
Squamous/metaplastic epithelium	34 (3.1)	4 (1.7)	26 (3.1)
Squamous/glandular/metaplastic epithelium	102 (9.3)	16 (7.0)	82 (9.8)
**Tests with abnormal or unusual cells (*n* = 31)**			
Atypical squamous cells of undetermined significance not ruling out high-grade lesion–ASC-H	7 (0.6)	2 (0.9)	5 (0.6)
Atypical, possibly non-neoplastic, squamous cells of undetermined significance–ASC-US	14 (1.3)	6 (2.6)	8 (1.0)
High-grade intraepithelial lesion (cervical intraepithelial neoplasia grade II and III)–HSIL	3 (3.3)	-	3 (3.4)
Low-grade intraepithelial lesion (cytopathic effect linked to HPV and cervical intraepithelial neoplasia grade II and III)–LSIL-H	2 (0.2)	1 (0.4)	1 (0.1)
Low-grade intraepithelial lesion (cytopathic effect linked to HPV and cervical intraepithelial neoplasia grade I)–LSIL	5 (5.5)	4 (1.7)	1 (0.1)

## Data Availability

The data presented in this study are available on reasonable request from the corresponding author. The data are not publicly available due to the possibility of identifying research participants in smaller rural locations.

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
