# Peer review of "Barriers to Access the Pap Smear Test for Cervical Cancer Screening in Rural Riverside Populations Covered by a Fluvial Primary Healthcare Team in the Amazon"

_ijerph, 2022, doi:10.3390/ijerph19074193_

Round 1

Reviewer 1 Report

In this manuscript titled “Barriers to access the Pap smear test for cervical cancer screening in rural riverside populations covered by a fluvial primary healthcare team in the Amazon”, the authors have provided a systematic evaluation of possible factors that influence the access to cervical cancer screening for women in rural riverside populations in the Amazon.

The manuscript is well written, with a detailed and informative introduction. The study is well-conducted and takes a variety of factors into consideration, and the results are very informative and presented in a manner to lay emphasis on each variable taken into consideration. The use of bivariate regression analysis and presentation of results along with the Odds ratios adds weight to their observations about the most crucial factors playing a role in the barriers towards access to screening. I must commend the authors for the Discussion section of the manuscript, which does a great job of bringing the results together and providing a detailed unbiased analysis of the entire study. The discussion also addresses the shortcomings, which is highly appreciated.

However, this is in sharp contrast to one major oversight by the authors, the missing tables 3 and 4 which are referred to in the text. This needs to be rectified before the manuscript can be considered fit for publication.

Major concerns:

The authors have referred to Tables 3 and 4 in the text but the associated tables are missing from the manuscript. Please fix.

Minor concerns:

In the few places in the Methods section, the methodology is unclear/confusing because of the way the sentences are framed. For example:

  • Lines 82-85: “The sample calculation considered the representativeness of the groups of interest in the larger project, that is, adults and elderly of both sexes, and children under the age of two years, and the probability of finding individuals from each group within each household.”. It is unclear what the authors mean by this. Was the presence of these groups of interest grounds/basis for selection of households for consideration in this study?
  • Lines 85-88: “The calculation considered a prevalence of 50% of the diseases of interest at 5% precision and 10% of possible losses or refusals, which were adjusted for the finite population, resulting in 239 households. The aim was to reach 731 people of the target population.” The first sentence is very confusing to read, and it is hard to understand what the authors mean by this. It is important that the authors make this part clearer or detailed because this is the most preliminary methodology of the entire study. Why was the aim to reach 731 people of the target population? I fail to understand the significance of this number.
  • Similarly, lines 108-109: “A descriptive analysis of the data was performed, including the four variables related to the outcome of interest according to the age groups <25 years and 25-59 years.” Please reframe to explain what this means.
  • In Table 1, please highlight (Bold/Italics/Underline/other means) the variable headers (Last screening test, reasons, results waiting time) to make the table easier to read.
  • Similarly, in Table 2, please highlight the variables under Structural determinants (years of education, occupation etc) and under Intermediary determinants (self-perception, level of difficulty etc) to make the table easier to read.
  • The authors have mentioned in the Discussion that “the retrieval of metaplastic or endocervical cells from the squamocolumnar junction is considered an indicator for the quality of smears”. It will be informative if the authors can comment on the quality of smears based on the sample information gathered from this study. For example : “601 samples (54.8%) had squamous/glandular epithelium; 34 samples (3.1%) had squamous/metaplastic epithelium; and 102 samples (9.3%) had squamous/glandular/metaplastic epithelium”.

Author Response

We thank the reviewers for their comments and/or suggestions, which helped to clarify this work. We have highlighted the changes within the document by using the track changes and also indicated them in the responses below.

---------

Reviewer 1

1) The manuscript is well written, with a detailed and informative introduction. The study is well-conducted and takes a variety of factors into consideration, and the results are very informative and presented in a manner to lay emphasis on each variable taken into consideration. The use of bivariate regression analysis and presentation of results along with the Odds ratios adds weight to their observations about the most crucial factors playing a role in the barriers towards access to screening. I must commend the authors for the Discussion section of the manuscript, which does a great job of bringing the results together and providing a detailed unbiased analysis of the entire study. The discussion also addresses the shortcomings, which is highly appreciated.

Answer: The authors are grateful for the reviewer's evaluation.

2) However, this is in sharp contrast to one major oversight by the authors, the missing tables 3 and 4 which are referred to in the text. This needs to be rectified before the manuscript can be considered fit for publication.

Answer: We checked the submitted version, which included the mentioned tables. Possibly there was some mistake during the manuscript journal editing, but now the tables have been reinserted in the revised version.

Major concerns

3) The authors have referred to Tables 3 and 4 in the text but the associated tables are missing from the manuscript. Please fix.

Answer: We checked the submitted version, which included the mentioned tables. Possibly there was some mistake during the manuscript journal editing, but now the tables have been reinserted in the revised version.

Minor concerns

In the few places in the Methods section, the methodology is unclear/confusing because of the way the sentences are framed. For example:

4) Lines 82-85: “The sample calculation considered the representativeness of the groups of interest in the larger project, that is, adults and elderly of both sexes, and children under the age of two years, and the probability of finding individuals from each group within each household.”. It is unclear what the authors mean by this. Was the presence of these groups of interest grounds/basis for selection of households for consideration in this study?

Answer: The study was part of a larger project, whose sampling was representative for all the groups mentioned. The number of selected households was based on the probability of finding dwellers of the interest groups, according to the average number of people from each group within each household, so that the calculated sample size could be reached (n=731 for all interest groups, n=221 for the population evaluated in the present study). As information about the other groups is dispensable, since the sample size calculation was performed for each one of the groups, and based on the reviewer's comment, the authors chose to simplify the description in the text to make it easier for readers to understand.

5) Lines 85-88: “The calculation considered a prevalence of 50% of the diseases of interest at 5% precision and 10% of possible losses or refusals, which were adjusted for the finite population, resulting in 239 households. The aim was to reach 731 people of the target population.” The first sentence is very confusing to read, and it is hard to understand what the authors mean by this. It is important that the authors make this part clearer or detailed because this is the most preliminary methodology of the entire study. Why was the aim to reach 731 people of the target population? I fail to understand the significance of this number.

Answer: 731 represents the total sample size, encompassing all interest groups evaluated in the larger study. Among these 731, 221 were women of interest to the study presented in the manuscript. We agree with the reviewer that by presenting the information in this way we were confusing readers and so we kept only the sample size information pertinent to the study in question, to make the text clearer.

6) Similarly, lines 108-109: “A descriptive analysis of the data was performed, including the four variables related to the outcome of interest according to the age groups <25 years and 25-59 years.” Please reframe to explain what this means.

Answer: We thank the reviewer for the comment. The text has been rewritten to make it clearer.

7) In Table 1, please highlight (Bold/Italics/Underline/other means) the variable headers (Last screening test, reasons, results waiting time) to make the table easier to read.

Answer: The headers were highlighted. The column text has also been realigned to the left, as it was before journal editing, to make the table easier to read.

8) Similarly, in Table 2, please highlight the variables under Structural determinants (years of education, occupation etc) and under Intermediary determinants (self-perception, level of difficulty etc) to make the table easier to read.

Answer: The variables names were highlighted. The column text has also been realigned to the left, as it was before journal editing, to make the table easier to read.

9) The authors have mentioned in the Discussion that “the retrieval of metaplastic or endocervical cells from the squamocolumnar junction is considered an indicator for the quality of smears”. It will be informative if the authors can comment on the quality of smears based on the sample information gathered from this study. For example : “601 samples (54.8%) had squamous/glandular epithelium; 34 samples (3.1%) had squamous/metaplastic epithelium; and 102 samples (9.3%) had squamous/glandular/metaplastic epithelium”.

Answer: The authors thank the reviewer again. This information is now presented in detail in Table 4, reinserted in the manuscript. The following paragraph of the Discussion section (starting at line 346) also refers to the quality of the smears, whose percentage of inadequacy identified in the study was 23.5%.

“In the present study, almost one quarter of the samples contained only one cell type (squamous or glandular), which is not a desirable quality indicator. This percentage is much higher than the recommended 5%, often requiring the repetition of the screening test [48,49].”

Reviewer 2 Report

da Silva et al. have performed a cross-sectional home-based survey regarding cervical cancer screening in rural riverside populations in the Amazon in the north region of Brazil. The administrative records of Pap smears performed in the territory from January 2016 to May 2019 were also analyzed in the study. Of the 221 women assessed, 8.1% had never undergone the test and 7.7% had undergone it more than three years ago. Multiparity, occupation in domestic activities, and lack of knowledge of the healthcare unit responsible for the service were associated with not undergoing the cervical cancer screening test. The findings revealed the existence of barriers for riverside women to access cervical cancer screening test. Comments In Brazil, the guidelines for cervical cancer screening recommend an annual screening test (Pap smear test) for women aged 25-64 years and an interval of three years after two consecutive negative tests. It is well documented that HPV-testing has proven superior to cytology with regards to sensitivity in detection of CIN and cancer. This should be included in the discussion. WHO recommends DNA testing as a first-choice screening method for cervical cancer prevention. Maybe Brazil should update and revise the guidelines for cervical cancer screening? DNA-based testing for human papillomavirus (HPV) has been shown to be more effective than today’s commonly used screening methods aimed at detecting and preventing cervical cancer, a major cause of death among women worldwide. https://www.euro.who.int/en/health-topics/noncommunicable-diseases/cancer/news/news/2021/9/who-recommends-dna-testing-as-a-first-choice-screening-method-for-cervical-cancer-prevention The American Cancer Society (ASC) recommends cervical cancer screening with an HPV test alone every 5 years for everyone with a cervix from age 25 until age 65. https://www.cancer.gov/news-events/cancer-currents-blog/2020/cervical-cancer-screening-hpv-test-guideline

I am really concerned about the quality of the Pap smears in Brazil. Administrative records showed a high percentage of unsatisfactory samples collected. 258 samples (23.5%) had insufficient cells (squamous or glandular epithelium found separately), indicating the need to repeat the screening test. This problem was associated with limited staff qualification. Where HPV tests are available as part of the national programme, HPV self-sampling offers an additional option to improve cervical cancer screening coverage. Vaginal self-sampling with HPV-DNA tests is a promising primary screening method for cervical cancer. Bansil, P., Wittet, S., Lim, J.L. et al. Acceptability of self-collection sampling for HPV-DNA testing in low-resource settings: a mixed methods approach. BMC Public Health 14, 596 (2014). https://doi.org/10.1186/1471-2458-14-596 Self-sampling can help reach a global target of 70% coverage of screening by 2030. Women may feel more comfortable taking their own samples, rather than going to see a health worker for cervical cancer screening. https://www.who.int/publications/i/item/WHO-SRH-2012 The clinical accuracy of hrHPV testing on a self-collected sample for detection of CIN3+ is high and supports its use as a primary screening test for all invited women. Inturrisi F, Aitken CA, Melchers WJG, van den Brule AJC, Molijn A, Hinrichs JWJ, Niesters HGM, Siebers AG, Schuurman R, Heideman DAM, de Kok IMCM, Bekkers RLM, van Kemenade FJ, Berkhof J. Clinical performance of high-risk HPV testing on self-samples versus clinician samples in routine primary HPV screening in the Netherlands: An observational study. Lancet Reg Health Eur. 2021 Nov 9;11:100235. doi: 10.1016/j.lanepe.2021.100235. PMID: 34918001; PMCID: PMC8642706. https://pubmed.ncbi.nlm.nih.gov/34918001/ Self-sampling for HPV-testing may be a valuable alternative for increasing cervical cancer screening coverage. Enerly E, Bonde J, Schee K, Pedersen H, Lönnberg S, Nygård M (2016) Self-Sampling for Human Papillomavirus Testing among Non-Attenders Increases Attendance to the Norwegian Cervical Cancer Screening Programme. PLoS ONE 11(4): e0151978. https://doi.org/10.1371/journal.pone.0151978 Self-sampling is important in cervical cancer screening as it has been shown to improve participation. Aranda Flores CE, Gomez Gutierrez G, Ortiz Leon JM, Cruz Rodriguez D, Sørbye SW. Self-collected versus clinician-collected cervical samples for the detection of HPV infections by 14-type DNA and 7-type mRNA tests. BMC Infect Dis. 2021 May 31;21(1):504. doi: 10.1186/s12879-021-06189-2. PMID: 34058992; PMCID: PMC8165795. https://pubmed.ncbi.nlm.nih.gov/34058992/ To eliminate cervical cancer, all countries must reach and maintain an incidence rate of below four per 100 000 women. Achieving that goal rests on three key pillars and their corresponding targets: Vaccination: 90% of girls fully vaccinated with the HPV vaccine by the age of 15; Screening: 70% of women screened using a high-performance test by the age of 35, and again by the age of 45; Treatment: 90% of women with pre-cancer treated and 90% of women with invasive cancer managed. Each country should meet the 90-70-90 targets by 2030 to get on the path to eliminate cervical cancer within the next century. https://www.who.int/initiatives/cervical-cancer-elimination-initiative In my experience, HPV-vaccination is more important than screening. In the manuscript there is no information about HPV vaccination in Brazil.

Minor revisions

Line 170-173, "Women who had more children, who reported domestic activities, and who were unable to inform or were unaware of the existence of a healthcare unit responsible for the community were less likely to have undergone a preventive screening test in the past three years (Table 3)." There is no Table 3 in the manuscript.

Line 185-186, "The joint assessment of these situations indicates that 140 screening tests (12.8%) should not have been performed (Table 4)." There is no Table 4 in the manuscript.

Author Response

We thank the reviewers for their comments and/or suggestions, which helped to clarify this work. We have highlighted the changes within the document by using the track changes and also indicated them in the responses below.

Reviewer 2

1)

In Brazil, the guidelines for cervical cancer screening recommend an annual screening test (Pap smear test) for women aged 25-64 years and an interval of three years after two consecutive negative tests. It is well documented that HPV-testing has proven superior to cytology with regards to sensitivity in detection of CIN and cancer. This should be included in the discussion. WHO recommends DNA testing as a first-choice screening method for cervical cancer prevention. Maybe Brazil should update and revise the guidelines for cervical cancer screening? DNA-based testing for human papillomavirus (HPV) has been shown to be more effective than today’s commonly used screening methods aimed at detecting and preventing cervical cancer, a major cause of death among women worldwide. https://www.euro.who.int/en/health-topics/noncommunicable-diseases/cancer/news/news/2021/9/who-recommends-dna-testing-as-a-first-choice-screening-method-for-cervical-cancer-prevention The American Cancer Society (ASC) recommends cervical cancer screening with an HPV test alone every 5 years for everyone with a cervix from age 25 until age 65. https://www.cancer.gov/news-events/cancer-currents-blog/2020/cervical-cancer-screening-hpv-test-guideline 

I am really concerned about the quality of the Pap smears in Brazil. Administrative records showed a high percentage of unsatisfactory samples collected. 258 samples (23.5%) had insufficient cells (squamous or glandular epithelium found separately), indicating the need to repeat the screening test. This problem was associated with limited staff qualification. Where HPV tests are available as part of the national programme, HPV self-sampling offers an additional option to improve cervical cancer screening coverage. Vaginal self-sampling with HPV-DNA tests is a promising primary screening method for cervical cancer. Bansil, P., Wittet, S., Lim, J.L. et al. Acceptability of self-collection sampling for HPV-DNA testing in low-resource settings: a mixed methods approach. BMC Public Health 14, 596 (2014). https://doi.org/10.1186/1471-2458-14-596 Self-sampling can help reach a global target of 70% coverage of screening by 2030. Women may feel more comfortable taking their own samples, rather than going to see a health worker for cervical cancer screening. https://www.who.int/publications/i/item/WHO-SRH-2012 The clinical accuracy of hrHPV testing on a self-collected sample for detection of CIN3+ is high and supports its use as a primary screening test for all invited women. Inturrisi F, Aitken CA, Melchers WJG, van den Brule AJC, Molijn A, Hinrichs JWJ, Niesters HGM, Siebers AG, Schuurman R, Heideman DAM, de Kok IMCM, Bekkers RLM, van Kemenade FJ, Berkhof J. Clinical performance of high-risk HPV testing on self-samples versus clinician samples in routine primary HPV screening in the Netherlands: An observational study. Lancet Reg Health Eur. 2021 Nov 9;11:100235. doi: 10.1016/j.lanepe.2021.100235. PMID: 34918001; PMCID: PMC8642706. https://pubmed.ncbi.nlm.nih.gov/34918001/ Self-sampling for HPV-testing may be a valuable alternative for increasing cervical cancer screening coverage. Enerly E, Bonde J, Schee K, Pedersen H, Lönnberg S, Nygård M (2016) Self-Sampling for Human Papillomavirus Testing among Non-Attenders Increases Attendance to the Norwegian Cervical Cancer Screening Programme. PLoS ONE 11(4): e0151978. https://doi.org/10.1371/journal.pone.0151978 Self-sampling is important in cervical cancer screening as it has been shown to improve participation. Aranda Flores CE, Gomez Gutierrez G, Ortiz Leon JM, Cruz Rodriguez D, Sørbye SW. Self-collected versus clinician-collected cervical samples for the detection of HPV infections by 14-type DNA and 7-type mRNA tests. BMC Infect Dis. 2021 May 31;21(1):504. doi: 10.1186/s12879-021-06189-2. PMID: 34058992; PMCID: PMC8165795. https://pubmed.ncbi.nlm.nih.gov/34058992/

Answer: We appreciate the reviewer's valuable contribution in improving the manuscript. Unfortunately, Brazil has not yet incorporated vaginal self-sampling into its public health practices, although several groups are already working to make this occur. Some colleagues have also conducted studies on this topic involving remote populations in the Amazon:

“Torres KL, Mariño JM, Pires Rocha DA, de Mello MB, de Melo Farah HH, Reis RDS, Alves VDCR, Gomes E, Martins TR, Soares AC, de Oliveira CM, Levi JE. Self-sampling coupled to the detection of HPV 16 and 18 E6 protein: A promising option for detection of cervical malignancies in remote areas. PLoS One. 2018 Jul 23;13(7):e0201262.

Rodrigues LL, Pilotto JH, Lima LR, Gaydos CA, Hardick J, Morgado MG, Martinelli KG, de Paula VS, Nicol AF. Self-collected versus clinician-collected samples for HSV-2 and HSV-2/HPV screening in HIV-infected and -uninfected women in the Tapajós region, Amazon, Brazil. Int J STD AIDS. 2019 Oct;30(11):1055-1062.”

Thus, this could not have been ignored in the study discussion. A paragraph was added to the Discussion section addressing the issue.

2) To eliminate cervical cancer, all countries must reach and maintain an incidence rate of below four per 100 000 women. Achieving that goal rests on three key pillars and their corresponding targets: Vaccination: 90% of girls fully vaccinated with the HPV vaccine by the age of 15; Screening: 70% of women screened using a high-performance test by the age of 35, and again by the age of 45; Treatment: 90% of women with pre-cancer treated and 90% of women with invasive cancer managed. Each country should meet the 90-70-90 targets by 2030 to get on the path to eliminate cervical cancer within the next century. https://www.who.int/initiatives/cervical-cancer-elimination-initiative In my experience, HPV-vaccination is more important than screening. In the manuscript there is no information about HPV vaccination in Brazil.

Answer: Once again, we thank the reviewer for his valuable contributions. A paragraph was added to the Discussion section addressing the mentioned issues.

Minor revisions

3) Line 170-173, "Women who had more children, who reported domestic activities, and who were unable to inform or were unaware of the existence of a healthcare unit responsible for the community were less likely to have undergone a preventive screening test in the past three years (Table 3)." There is no Table 3 in the manuscript.

Answer: We checked the submitted version, which included the mentioned tables. Possibly there was some mistake during the manuscript journal editing, but now Table 3 have been reinserted in the revised version.

4) Line 185-186, "The joint assessment of these situations indicates that 140 screening tests (12.8%) should not have been performed (Table 4)." There is no Table 4 in the manuscript.

Answer: We checked the submitted version, which included the mentioned tables. Possibly there was some mistake during the manuscript journal editing, but now Table 4 have been reinserted in the revised version.

Round 2

Reviewer 1 Report

The authors have satisfactorily replied to all the concerns and have made all the relevant suggestions in their revised manuscript. I appreciate that they responded to all the comments and I support the publication of this manuscript.